# Cyclic orthogonal convolutions for long-range integration of features

**Federica Freddi**[*]        **Jezabel R Garcia**[*]        **Michael Bromberg**        **Sepehr Jalali**

**Da-Shan Shiu**            **Alvin Chua**            **Alberto Bernacchia**

MediaTek Research UK
{federica.freddi, jezabel.garcia, michael.bromberg, sepehr.jalali,
ds.shiu, alvin.chua, alberto.bernacchia}@mtkresearch.com

## Abstract

In Convolutional Neural Networks (CNNs) information flows across a small neighbourhood of each pixel of an image, preventing long-range integration of features before reaching deep layers in the network. Inspired by the neurons of the human visual cortex responding to similar but distant visual features, we propose a novel architecture that allows efficient information flow between features $z$ and locations $(x, y)$ across the entire image with a small number of layers. This architecture uses a cycle of three orthogonal convolutions, not only in $(x, y)$ coordinates, but also in $(x, z)$ and $(y, z)$ coordinates. We stack a sequence of such cycles to obtain our deep network, named CycleNet. When compared to CNNs of similar size, our model obtains competitive results at image classification on CIFAR-10 and ImageNet datasets. We hypothesise that long-range integration favours recognition of objects by shape rather than texture, and we show that CycleNet transfers better than CNNs to stylised images. On the Pathfinder challenge, where integration of distant features is crucial, CycleNet outperforms CNNs by a large margin. Code has been made available at: https://github.com/netX21/Submission

## 1  Introduction

Several computer vision tasks require capturing long-range dependencies between features. For example, in order to recognize a teapot, it is necessary to identify and integrate several of its parts in their correct spatial relationship, e.g. a spout, a handle, a lid, while each part by itself is usually not enough to determine the object class. Convolutional layers have proven very useful at several computer vision tasks such as classification, and they are at the core of most state-of-the-art neural networks (LeCun et al. (2015)). A convolution transforms each pixel depending on a few neighboring pixels, and the transformation is shared across all pixels; however, the small size of the neighbourhood implies that in a single layer features at distant locations cannot be integrated. In deep convolutional networks, the receptive field size increases sub-linearly with depth (Luo et al. (2017)), therefore a large number of layers is necessary for integrating features across long distances. We propose a novel neural network architecture that retains the efficient parameterization of convolutions, while promoting long-range interactions of distant features.

Similar to the human visual system, individual 'neurons' within each CNN have receptive fields with size increasing with the number of layers. However, while the human visual system achieves large

---

[*]equal contribution

3rd Workshop on Shared Visual Representations in Human and Machine Intelligence (SVRHM 2021) of the Neural Information Processing Systems (NeurIPS) conference, Virtual.

receptive fields within a handful of layers, CNNs need many more (Luo et al. (2017)). In fact, the number of layers keeps expanding to 100s or 1000s of layers for state of the art CNN models (Tan & Le (2019); He et al. (2016)). A neuron of the visual system connects not only to other neurons responding to similar spatial locations, but also to neurons at distant locations, provided that they share similar visual features, such as edge orientation (see Fig.1a). These connections not only enlarge the receptive fields, but they also facilitate solving tasks that require long-range integration of features. For example, they allow tracking curved contours (Fig.1b). The neurons responsible for the long-range connections are excitatory pyramidal cells, which are known to have a long range connections in the cortex. Instead, inhibitory GABA-ergic cells do not have this property.

Our model architecture agrees with these biological principles of information processing taking place in the brain. Given a tensor of horizontal ($x$) and vertical ($y$) coordinates and features ($z$), we propose to apply convolutions not only on ($x, y$) coordinates, but also on ($x, z$) and ($y, z$) in sequence. These orthogonal convolutions allow the interaction of 'neurons' at distant locations, provided that they share similar visual features, similarly to the human visual system. We show that, after a cycle of three convolutions over the three axes, each element of the tensor depends on all elements of the input, and features at all locations have a chance to interact. Our model, named CycleNet, is obtained by adding a permutation of the axes to a convolution, and therefore is easy to compare with a standard CNN. In order to evaluate our model, we compare it with a CNN baseline network with identical architecture, i.e. same number of parameters and tensor shape in all layers.

In this paper we show that Cyclenet: outperforms the CNN baseline at image classification on CIFAR-10 and ImageNet (Krizhevsky (2009), Deng et al. (2009)), and approaches the performance of ResNet and MobileNet (He et al. (2016), Howard et al. (2017)) with a similar number of parameters; achieves the maximum receptive field size after a single cycle and outperforms the baseline CNN in transfer learning to stylised data, suggesting the use of more features dependent on shape rather than texture; outperforms the baseline by a large margin at the Pathfinder challenge, a task inspired by the study of human visual perception where CNNs are known to fail dramatically (Linsley et al. (2019)).

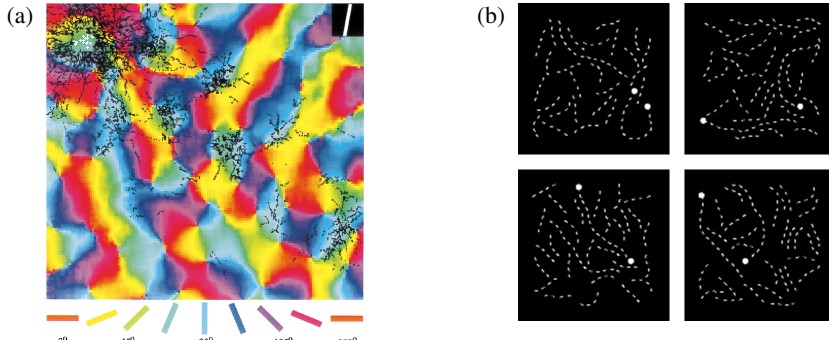

Figure 1: (a) The image shows a patch of visual cortex, different colors denote features to which neurons respond to (oriented bars). Black dots denote connections made by a few neurons, denoted by white crosses. Such neurons respond to green/blue features, and tend to connect with other neurons at nearby spatial locations, for any feature, but also to neurons at distant locations that respond to the same features. Adapted from Bosking et al. (1997) (b) Pathfinder challenge (Linsley et al. (2019)). Each image has two circles attached to a path. The goal of this task is to classify images into connected and disconnected, two examples shown for each class.

## 1.1 Related work

A few recent papers showed that deep convolutional networks struggle to learn features integrating large spatial domains. For instance Luo et al. (2017) showed that the receptive field size of a deep CNN is smaller than the sum of its kernel sizes. Geirhos et al. (2019) showed that ImageNet-trained ResNet relies on small image patches and textures, rather than object shapes, and does not transfer well to stylised images. Linsley et al. (2019) showed that ResNet struggles on the Pathfinder challenge, a simple task that requires integrating features over long distances.

Other methods have been proposed to learn large scale features. Pooling layers increase the receptive field size in CNNs, but they lose a significant amount of information, including the spatial relationship

between features (Boureau et al. (2010)). Multiscale pyramids use kernels of different sizes arranged in parallel, but larger kernels have lower resolution in order to keep a reasonable number of parameters (Farabet et al. (2013)). Deformable convolutions adaptively learn the shape of the kernel, at the cost of an increased complexity of the model (Dai et al. (2017)). Dilated convolutions increase the kernel size along network depth (Yu & Koltun (2015)) and they are equivalent to a tensor decomposition (Huszar (2016)). Tensor decompositions have been widely used to reduce the complexity of convolutional and dense layers (Kuzmin et al. (2019); Novikov et al. (2015)). We show below that one cycle of CycleNet is equivalent to a decomposition of a dense layer, in the simple case of $1 \times 1$ convolutions and linear activations, but not in the general case. CycleNet is also different from a 3D convolution (Ji et al. (2013)), since spatial coordinates are fully connected in the second and third layers of a cycle.

Among non-convolutional architectures, self-attention networks naturally capture long-range dependencies. Initially designed for natural language processing (Vaswani et al. (2017)), transformers were recently shown to exhibit good performance on vision tasks (Wang et al. (2017)). Similar to self-attention networks, CycleNet breaks translational symmetry and has the best performance on ImageNet when standard convolutions are included in the first layers (Ramachandran et al. (2019)).

## 2 Model: A cycle of three orthogonal convolutions

In this section we describe a single cycle, the basic building block of CycleNet, which is composed of three layers arranged in sequence (Fig.2a). We start by describing a single convolutional layer. We denote the input tensor as $I(x, y, z)$, at coordinates $x$ (horizontal), $y$ (vertical) and feature $z$. The output tensor $S$ is equal to:

$$S(x, y, z) = \sum_{dx, dy, z'} K(dx, dy, z, z') I(x + dx, y + dy, z'). \tag{1}$$

where $K$ denotes the convolutional kernel. We assume zero padding and kernel stride equal to one, while other parameters vary in different experiments (kernel size, input and output tensor shapes). The convolution operation is local: the output at a given location $(x, y)$ depends only on the input displaced by $dx, dy$, spanning neighboring pixels up to the kernel size, which is typically much smaller than the size of the input. On the other hand, features $z$ are all-to-all fully connected. This is a standard convolutional layer and is depicted as the first layer of Fig.2a (green).

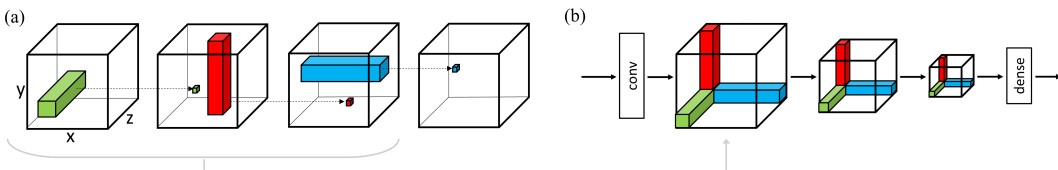

Figure 2: Illustration of CycleNet. (a) One cycle is defined as three convolutions performed in sequence: first on $(x, y)$ (green), second on $(x, z)$ (red), and third on $(y, z)$ (blue), each convolution is followed by BatchNorm, ReLU and Dropout (not shown). (b) The full model consists of a stack of cycles, plus a first convolutional layer and a final dense layer.

Convolutional layers are usually stacked and interleaved with other types of layers, such as nonlinearities, down(up)sampling, skip connections, normalization, regularization, etc. In this study, we use BatchNorm (Ioffe & Szegedy (2015)), ReLU (Glorot et al. (2011)), Dropout (Srivastava et al. (2014)), and no skip connections. In order to promote long-distance integration, we propose to apply convolutions not only in the $(x, y)$ plane, but also in the $(x, z)$ and $(y, z)$ planes. With some abuse of notation, we denote $I$ as the output of the previous layer, and $S$ as the output of the current layer. The second layer in a cycle is given by the following convolution,

$$S(x, y, z) = \sum_{dx, y', dz} K(dx, dz, y, y') I(x + dx, y', z + dz), \tag{2}$$

also illustrated in Fig.2a (red). Here, vertical coordinates $y$ are fully connected. Subsequently, after another BatchNorm, ReLU and Dropout, we apply a convolution in the $(y, z)$ plane:

$$S(x, y, z) = \sum_{x', dy, dz} K(dy, dz, x, x') I(x', y + dy, z + dz). \tag{3}$$

This is the third and final layer in a cycle, illustrated in Fig.2a (blue), and is also followed by BatchNorm, ReLU and Dropout. Here, horizontal coordinates $x$ are fully connected. The sequence of three convolutions along the three different axes constitutes a cycle. Note that a cycle could be defined with a different ordering of the three convolutions, but we did not explore other configurations.

Further architecture details about the long range integration, the number of parameters, the comparison with a convolutional baseline and the scaling to deeper networks can be found in Appendix A.

## 3 Experiments

### 3.1 Receptive fields

We train CycleNet and the baseline CNN on CIFAR-10 and ImageNet, benchmarks for image classification over which CNNs have been extensively optimized. Experimental results can be seen in Figure 3 and detailed description and discussion of these experiments can be found in appendix B.1. Figure 3 shows that CycleNet has a higher accuracy than the baseline and is comparable to state of the art CNN models of similar size. Is this because CycleNet learns substantially different representations of the images compared to standard CNN? We investigate this using receptive fields.

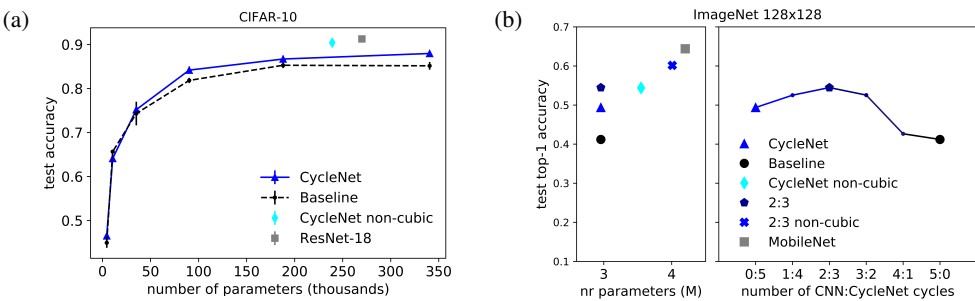

Figure 3: Test accuracy on image classification. (a) Performance of CycleNet and the CNN baseline on CIFAR-10, as a function of the number of parameters. Each point corresponds to a different network depth, from one to six cycles, and is an average of five experiments with identical optimized hyperparameters. ResNet-18 is taken from (He et al. (2016)). (b). Left: performance of CycleNet, the CNN baseline and 2:3 hybrid model on ImageNet ($128 \times 128$) as a function of the number of parameters. MobileNet is taken from (Howard et al. (2017)). Right: we progressively substitute CycleNet with standard convolutional cycles. Non-cubic: experiments with non-cubic tensor shape. Details about different CycleNet architectures an be found in appendix B.1.

At the end of a cycle, each element of the output tensor may depend on all elements of the input tensor (see appendix A.1), thus CycleNet is expected to have large receptive fields. In contrast, the receptive fields of CNNs are usually small (Luo et al. (2017)). Here, we compute the receptive fields at the output of each cycle of our models with 18 layers for CIFAR and 15 layers for ImageNet: first, we compute saliency maps following Simonyan et al. (2014), then we use those maps to compute receptive fields following Luo et al. (2017). Details can be found in section 3.3.

Figure 4a shows receptive field size as a function of depth, where the receptive field is normalized by the resolution (32 for CIFAR, 128 for ImageNet). Each bar is computed across 100 receptive fields sampled at random, 10 activations times 10 images. As expected, CycleNet achieves a large receptive field size after one cycle (3 layers), while the CNN baseline increases the receptive field size with depth, and approaches the size of CycleNet after 18 layers. Fig.4b shows a few visual examples of receptive field shapes for both CycleNet and the CNN baseline at different layer positions.

Larger receptive fields are fundamental in tasks requiring long-range integration of features and the features developed implicitly rely on more global patters, favouring classification of images by shape rather than texture. These are investigated in Section 3.2 and section 3.3.

(a)

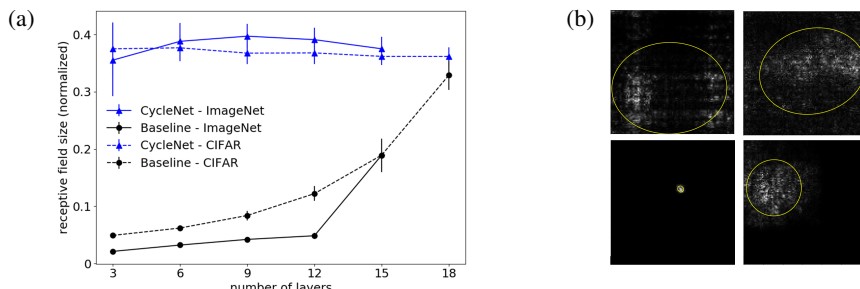

(b)

Figure 4: Receptive field sizes and shapes. (a) Normalized receptive field size as a function of activation depth (number of layers) of CycleNet and the CNN baseline, for ImageNet and CIFAR-10. Each point is the mean and standard deviation computed on 100 receptive fields chosen at random. CycleNet achieves large size after one cycle (3 layers), while the size increases with depth in the baseline, and achieves similar size after 18 layers. (b) Sample saliency maps in CycleNet (top) and the baseline (bottom) on ImageNet, after 3 (left) and 15 layers (right). Yellow ellipses show the receptive field shape, covering 3 standard deviations.

## 3.2 Pathfinder challenge

In order to test whether large receptive fields help integrating long-range features, we evaluated CycleNet on the Pathfinder challenge (Fig.1d). This task is inspired by the study of human visual perception, and requires tracking paths over long distances to distinguish whether two white circles are connected. Deep convolutional networks struggle on this task (e.g. ResNet50), because of their limited ability of integrating features over long distances (Linsley et al. (2019)). Since CycleNet integrates all pixels after one cycle, we predict that it should perform well on this task. Linsley et al. (2019) proposes a biologically-inspired horizontal gated recurrent neural network that performs ~100% on the task, but their model is hard to compare with standard CNNs. Details about the Pathfinder dataset as well as the architectures used can be found in appendix B.3.

Figure 5 shows the performance of CycleNet and the CNN baseline in 3 challenges of increasing difficulty, i.e. increasing path lengths: $n = 6$, $9$, and $14$. For kernel sizes $4, 8$, and $12$, the CNN baseline does not learn the task. It performs at chance level ($50\%$), except in one experiment of path length 6. On the other hand, CycleNet shows good accuracy, up to nearly $100\%$, suggesting that it is able to take advantage of its larger receptive fields. Furthermore, its performance decreases with path length and increases with kernel size. Note that large receptive fields may not be enough to solve this task, it also requires an increasing level of expressivity at larger path lengths. This is provided by a larger kernel size in CycleNet. For kernel size 20 the CNN baseline is able to perform the task, sometimes better than CycleNet; note that however for this kernel size the training has not converged as we were limited by computational resources and the models were very large due to the size of the kernel. We can only observe that for this case the CNN converged faster. At this size, a single kernel covers a substantial part of the image and is effectively long-range. However, the number of parameters is large in this case and the efficiency of the parameterization is lost.

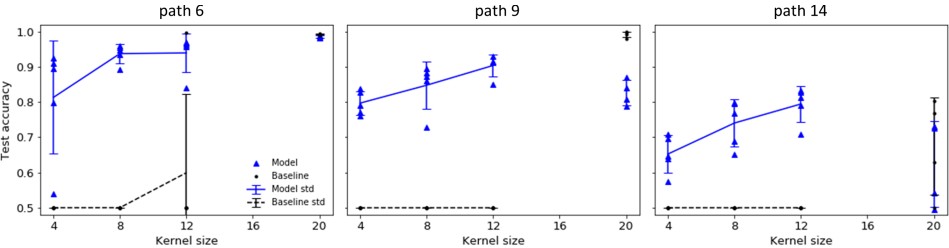

Figure 5: Pathfinder challenge. Each panel shows the accuracy of CycleNet and the CNN baseline as a function of kernel size. Task difficulty increases from left to right (path length $n = 6, 9, 14$). Each point is one experiment, the mean and standard deviation across multiple experiments is shown by error bars. For any kernel sizes $4, 8, 12$, the baseline does not learn the task (chance $= 0.5$), while CycleNet shows a good accuracy that increases with kernel size and task ease. For learning the task the CNN needs to use a kernel size of 20, much larger than commonly used CNN's kernels.

### 3.3 Stylised images

Given that CycleNet has large receptive fields, we hypothesise that it should classify images by the shape rather than texture of objects. Geirhos et al. (2019) designed new benchmarks to explore the shape vs texture bias of deep neural networks trained on ImageNet (IN). In the Stylised ImageNet dataset (SIN), the texture of each image is replaced with a randomly selected painting style (Fig.6, top). In the Cue-Conflict dataset (CC), a few images are generated by iterative style transfer from selected textures consisting of patches of other classes (Fig.6, bottom). We predict that our IN pre-trained CycleNet should transfer better than the baseline to both SIN and CC. We generate SIN following the scripts at https://github.com/rgeirhos/Stylized-ImageNet, and data for CC is sourced from https://github.com/rgeirhos/texture-vs-shape. Both experiments are evaluated at resolution 128x128.

Table 1 shows evaluation top-5 accuracy on Stylised ImageNet and top-1 accuracy on the Cue-Conflict dataset. The results confirm that CycleNet transfers better to both datasets. Although the overall accuracy on ImageNet is higher for a hybrid network where the first two cycles have standard convolutions (2:3), the best transfer to stylised images is obtained by CycleNet, where all cycles use orthogonal convolutions (0:5). These results confirm that large receptive fields are instrumental for biasing a neural network towards shape, and that the base performance on ImageNet by itself is not a good predictor of such bias.

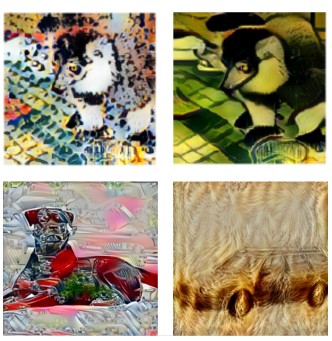

Figure 6: Stylised ImageNet (top) and Cue-Conflict dataset (bottom).

| cycles | IN→IN | IN→SIN | IN→CC |
|--------|-------|--------|-------|
| 0:5 | 49.4 | **8.8** | **16.4** |
| 1:4 | 52.5 | 7.0 | 14.0 |
| 2:3 | **54.5** | 6.3 | 14.7 |
| 3:2 | 52.5 | 4.8 | 12.8 |
| 4:1 | 42.6 | 3.6 | 13.5 |
| 5:0 | 41.2 | 3.4 | 11.4 |

Table 1: Accuracy values (%) for the CNN:CycleNet models trained on IN and evaluated on SIN and CC datasets (at $128 \times 128$ resolution). As we progressively substitute CycleNet with CNN cycles, the ability of the network to transfer to stylised datasets reduces. Instead, the base accuracy on IN is highest when the first two cycles are CNN (model 2:3).

## 4   Discussion

We proposed a biologically inspired neural network architecture, CycleNet, that achieves better performance than a CNN baseline in classification tasks, and it develops a significantly different representation of the input image, where node activations have large receptive fields and are thus able to represent large portions of an image. This is in agreement with the human visual system that achieves large receptive fields within few layers and in sharp contrast with CNNs, which have smaller receptive fields. CycleNet transfers better than the CNN baseline to stylised images, suggesting that the large receptive fields bias the model towards shape rather than texture of objects, and performs much better at the Pathfinder task, which requires long-range integration of features.

We emphasize that CycleNet loses translational symmetry, which is considered a strong feature of CNNs. However, recent evidence suggests that this property may not be crucial. Symmetries do not have to be hard coded in the architecture: they can be learned by stochastic gradient descent (Achille & Soatto (2018)) and data augmentation (Taylor & Nitschke (2017)). Furthermore, several non-translational symmetric architectures recently achieved near state-of-the-art performance in image classification (Ramachandran et al. (2019)). The good performance of CycleNet on classification adds to this line of research, suggesting that built-in translational symmetry may not be necessary.

Performance in other tasks is likely to benefit from long-range integration, such as image segmentation, generation, reconstruction, etc. Future work may focus on testing CycleNet on those tasks.

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

# A Appendix: Architecture details

## A.1 Long-range integration

We combine Eqs.1, 2 and 3 to express the effect of one cycle. We consider the simple case of $1 \times 1$ convolutions (thus $dx = dy = dz = 0$), and we ignore all nonlinearities in between layers, obtaining

$$S(x, y, z) = \sum_{x', y', z'} K_3(x, x') K_2(y, y') K_1(z, z') I(x', y', z'). \tag{4}$$

where $K_1$, $K_2$ and $K_3$ are the three successive kernels. Therefore, after a cycle of three convolutions, each element of the output tensor depends on all elements of the input tensor. It is straightforward to check that, even for $k \times k$ convolutions ($k > 1$) and nonlinearities, each element of the output tensor may still depend on all elements of the input tensor. However, it is not guaranteed that CycleNet makes use of the entire receptive range after training on a given task, and therefore we test this hypothesis in Section 3.1.

Note that if we substitute

$$K(x, y, z, x', y', z') = K_3(x, x') K_2(y, y') K_1(z, z') \tag{5}$$

into Eq.4, then the expression becomes equivalent to a dense layer, where the kernel is decomposed into three factors. Thus, Eq.4 is equivalent to a tensor decomposition. However, CycleNet is different from a tensor decomposition when nonlinearities are included between layers, and in the more general case for convolutions of kernel size $k > 1$.

## A.2 Number of parameters and operations

The expression to calculate the number of parameters in CycleNet is the same as in a standard convolutional layer, simply the convolutions are performed along a different axis. We denote by $k$ the size of the kernel, by $X_{in}, Y_{in}, Z_{in}$ the shape of the input tensor and by $X_{out}, Y_{out}, Z_{out}$ the shape of the output tensor. The number of parameters of the first layer is $k^2 Z_{in} Z_{out}$, each one operated on $X_{out} Y_{out}$ times. Similarly, the number of parameters of the second layer is $k^2 Y_{in} Y_{out}$, times $X_{out} Z_{out}$ operations, and for the third $k^2 X_{in} X_{out}$ parameters, times $Y_{out} Z_{out}$ operations. Note that the number of parameters of CycleNet may differ from a standard convolution if the number of horizontal pixels $X$ or vertical pixels $Y$ is different from the number of features $Z$.

## A.3 Cubic baseline

In order to compare the performance of CycleNet with standard CNNs, we define a baseline to study the relative improvement introduced by orthogonal convolutions. The baseline is exactly identical to CycleNet, except that we do not permute the axes, instead we always perform convolutions along the same axis. This corresponds to a standard stack of convolutional layers, interleaved with BatchNorm, ReLU and Dropout. However, in order to keep the number of parameters equal between CycleNet and the CNN baseline, we consider cubic tensors in most of our experiments, i.e. in which the number of horizontal pixels $X$, vertical pixels $Y$ and the number of features $Z$ are equal within each cycle. This is a strong constraint that makes such baseline quite specific, but we still refer to it simply as "baseline". It is likely that a better performance could be obtained, both by CycleNet and the CNN, without this cubic constraint. Therefore, we also perform experiments on CycleNet with non-cubic tensors, and compare its performance with other convolutional architectures, e.g. ResNet and MobileNet, which are also non-cubic.

## A.4 Deep network

As illustrated in Fig.2b, we stack cycles in sequence to construct a deep network model. We add a standard convolution as a first layer of the network, to obtain the appropriate number of features to be used in the first cycle, and a dense layer as the last layer, thus reducing the (flattened) tensor to the size of the final output. In cubic models, tensor width is changed from one cycle to the next using tri-linear interpolation. In non-cubic models, tensor shape is changed by controlling the output size of the fully connected coordinate at each layer. In the case of the Pathfinder challenge, we use global pooling before the dense layer, in order to match the baseline architectures of Linsley et al. (2019). Code will be made available to the reviewers in the supplementary material, and to the public upon acceptance of the paper.

# B  Appendix: Experimental details

## B.1  Image classification

We start by training CycleNet on CIFAR-10 and ImageNet (at $128 \times 128$ resolution), standard benchmarks for image classification on which CNNs have been extensively optimized (Fig.1a,b). We do not aim at beating the current state-of-the-art. Our goal is to compare CycleNet with standard CNNs of equal or similar size, and use them to test our working hypotheses in the next sections. We use relatively small models for ease of implementation, in order to explore a variety of hyperparameters. The CIFAR dataset consists of 60,000 32x32 colour images in 10 classes, 50,000 training images and 10,000 test images. We use the standard cross-entropy loss with L2 regularization. We train on a single GPU (GeForce RTX 2080 Ti) using the Keras API in TensorFlow. The training cycle is 300 epochs and the network hyperparameters are optimized by running twelve experiments varying dropout rates and L2 lambda. RMSprop was used with batch size 64 and initial learning rate 0.001, which is divided by 10 when the error plateaus. Data augmentation is implemented with $20°$ rotation range, up to 20% shift of height and width, and horizontal flip on 50% of the data. All convolutions have kernel size $k = 3$. The first convolution outputs 80 features, and the deepest architecture has the following tensor widths in successive cycles: 75, 60, 45, 30, 15, 5. For the shallower networks, we use the smaller end of the sequence, e.g. width 5 for one cycle, widths 15, 5 for two cycles, widths 30, 15, 5 for three cycles and so on and so forth. The non-cubic CycleNet starts with a convolution with 100 features ($32 \times 32 \times 100$) followed by four cycles ($45 \times 45 \times 100$, $30 \times 30 \times 66$, $15 \times 15 \times 33$, $5 \times 5 \times 8$). Fig.3a shows the performance of models of up to six cycles (18 layers, $k = 3$) on CIFAR-10. CycleNet performs better than the CNN baseline, and the non-cubic CycleNet approaches the performance of a ResNet-18 (He et al. (2016)) with a similar number of parameters. These results suggest that long-range integration may be instrumental for image classification, but their significance is limited by the small resolution of CIFAR-10 ($32 \times 32$).

To confirm our results on higher resolution, we next tested CycleNet on ImageNet ($128 \times 128$). Fig.3b (left) shows the performance of five-cycle models (15 layers, $k = 3$) on ImageNet. The ImageNet dataset consists of 1.28 million training and 50,000 test images in 1000 classes. We use $128 \times 128$ resultion, and the standard cross-entropy loss. We train on a 24 GPUs (GeForce RTX 2080 Ti) distributed training system (Sergeev & Del Balso (2018)) for 120 epochs. Adam optimizer was used with effective batch size 384 and learning rate 0.024 (using warm-up as in (Goyal et al. (2017)), which is divided by 10 when the error plateaus. Similar to CIFAR, data augmentation is implemented with $20°$ rotation range, up to 20% shift of height and width, and horizontal flip on 50% of the data. All convolutions have kernel size $k = 3$. In the cubic models, the first convolution outputs 128 features, followed by cycles of the following tensor widths: 106, 106, 106, 106, 12. The non-cubic CycleNet starts with a convolution with 260 features ($128 \times 128 \times 260$) followed by five cycles ($106 \times 106 \times 260$, $106 \times 106 \times 190$, $106 \times 106 \times 159$, $106 \times 106 \times 159$, $10 \times 10 \times 11$). The non-cubic 2:3 CNN:CycleNet model has a convolution of 200 filters, followed by two CNN cycles (equivalent to 6 convolutions of output features $200, 200, 200, 190, 170, 159$) and three CycleNet cycles ($106 \times 106 \times 159$, $106 \times 106 \times 159$, $10 \times 10 \times 11$).

CycleNet performs better than the baseline, but its accuracy is significantly lower when compared to MobileNet (Howard et al. (2017)), which has a slightly larger size. We experiment with larger models, and we followed two approaches to increase performance up to a comparable level. First, similar to our CIFAR-10 experiment, we try a non-cubic architecture and find a 10% gain in relative performance. Second, following previous work (Ramachandran et al. (2019)), we hypothesise that image classification benefits from standard convolutions in the first layers, therefore we vary the number of initial cycles with standard convolutions. Fig.3b (right) shows that a model with two cycles of standard convolutions (2:3) has the best performance, and combining this with a non-cubic architecture, the model approaches the performance of MobileNet with a similar number of parameters. This motivates using CycleNet and the CNN baseline for testing our working hypotheses in the next sections.

We used relatively small CycleNet models, in order to have a baseline that can be easily compared with, and to be able to explore CycleNet with a variety of choices of hyperparameters. It would be interesting to look at whether state of the art performance can be achieved on datasets of higher resolution and larger models, for example by introducing residual connections. This would however require significantly more computational resources.

## B.2 Calculation of the receptive field

To calculate the receptive field, for a given node in the network and a given input image, we compute the gradient of the activation of that node with respect to the image. The saliency map is a grayscale image, obtained by taking the absolute value of the gradient for each pixel and summing across the three color channels (Simonyan et al. (2014)). The saliency map is normalized, such that the sum across pixels is one, and is interpreted as a 2-dimensional probability density. The receptive field size is computed by the square root of the total variance (the trace of the covariance matrix) of that density. We use the total variance since it quantifies the scatter of the saliency map along both axes independently. The receptive field shape is defined as the ellipsoid containing 3 standard deviations on both axes.

## B.3 The Pathfinder challenge

The Pathfinder dataset is composed of $900,000$ training and $100,000$ test images, and has two classes. We use the standard cross-entropy loss with L2 regularization. In our experiment, in addition to the first convolution, CycleNet has a single cycle at $128 \times 128 \times 128$ features, global pooling (as in Linsley et al. (2019)) and a dense layer. The experiments run on a single GPU (GeForce RTX 2080 Ti), Adam optimizer was used with learning rate $0.001$. We use $5$ times more distractors than the base dataset.

We generate three datasets of increasing difficulty, for different path lengths: $n = 6, 9$, and $14$, at $128 \times 128$ resolution, using the code available at https://github.com/drewlinsley/pathfinder. We study performance of one-cycle models as a function of kernel size, $k = 4, 8, 12$ and $20$.

For any kernel sizes $4, 8, 12$, the baseline does not learn the task (chance $= 0.5$), while CycleNet shows a good accuracy that increases with kernel size and task ease (see Figure 5). For kernel size $20$, which allows convolutions to integrate long-range, the CNN is able to learn the task. However, for this kernel size, the accuracy values reported here are not final because the models had not converged. The training stopped before convergence due to limitations in computational resources.

Note that CNNs typically have kernel size smaller than $10$, since the number of parameters becomes substantial for larger kernels.

