# OpenReview forum: "Cyclic orthogonal convolutions for long-range integration of features"
_NeurIPS.cc/2021/Workshop/SVRHM — SVRHM 2021 Poster_

### Official Review · Reviewer_VJoG · 2021-10-29
**Interesting idea but not quite there yet**

**Rating:** 4
**Confidence:** 4

**Review:**


The paper proposes cyclic orthogonal convolutions as a means to grow receptive fields fast in CNNs. The authors show a small improvement of their cyclic convolution model over a simple CNN baseline on CIFAR-10, ImageNet and Stylized ImageNet. Overall it's an interesting idea, but not executed very convincingly.

The biological motivation is weak at best. Long-range horizontal connections in cortex, which the authors use as motivation, are feature specific, i.e. between corresponding orientation domains. In contrast, in the authors' setup, they interact across all features. Moreover, the long-range connections are only along the x and y direction, but not in oblique directions. In my opinion, the author's proposal is not closer to biology than vision transformers, which also provide an all-to-all spatial interaction, albeit with a different mechanism and arguably stronger performance on large-scale datasets.

The experiments with simple CNNs are nice and show a trend in the right direction, but in order to show that the cyclic convolutions are also of practical use, more extensive and competitive results would be necessary. The authors argue that also in ResNets receptive fields grow sublinearly with depth. If that's the case, why don't they show that incorporating cyclic orthogonal convolutions improves a standard ResNet-50 model on ImageNet?

I don't find the pathfinder results very convincing. It has been shown before (this workshop, last year) that CNNs can also learn Pathfinder once the training setup is slightly adjusted [1]


[1] https://openreview.net/forum?id=dPwyQnHUVvw

---

> ### Author Response · Authors · 2021-12-06
> **Oblique connections**
>
> We thank the reviewer for the feedback. We would like to point out that for a single orthogonal convolution we only have a feature-specific kernel, or at least limited to a features neighbourhood defined by the kernel size. Therefore, over a single orthogonal convolutional layer, not all features interact with each other. Only as the convolutions over different planes are stacked on top of each other to form a cycle, we introduce interaction between all features and spatial locations. We agree with the reviewer that there are no direct oblique connections, but the interactions build also in oblique directions after an entire cycle.

---

### Official Review · Reviewer_Bgcs · 2021-10-31
**A clearly motivated approach with some interesting findings**

**Rating:** 9
**Confidence:** 4

**Review:**

This paper attempts to enable CNNs to learn long range spatial dependencies, typically only possible at great depth, in the early layers. To acheive this the authors propose CycleNet, a network of 'cycles' of orthogonal convolutions. These convolutions are performed across the three coordinate planes and have a subtaintially larger receptive field than a typical convolution without a dramatic increase in the number of parameters. The motivation for this work is comprehensive and the architecture is well described and intuitive. Experimental results show that CycleNet significantly improves performance over a baseline on the pathfinder challenge and also provides a modest improvement / increase in parameter efficiency on CIFAR-10.

The authors touch on a biological basis for the ideas explored here but I feel that the full potential of this line of reasoning is not realised. For example, the authors show improved generalisation to stylised ImageNet only in an Appendix when it is arguably among the most exciting results of the paper. These could be further augmented with the addition of other biological similarity measures such as the brain-score (https://www.brain-score.org/). Finally, it would be valuable for the authors to delve deeper into the related biology, perhaps identifying specific cell types or psychophysical results that they feel are better represented by the CycleNet model.

Overall, this is a well presented and clearly motivated work with promising results, a strong accept.

---

> ### Author Response · Authors · 2021-12-06
> **Stylized ImageNet, biological background and future work**
>
> We thank the reviewer for the useful and actionable feedback. We will make sure to introduce the Stylized ImageNet back in the main body of the paper. Thank you for the suggestions regarding the brain score, this is exciting and we will look into this as part of future work. Related to biology we refer in particular to excitatory pyramidal cells, which are known to have long-range connections in the cortex. As opposed, for example to inhibitory GABA-ergic cells. We will add details in the updated paper version

---

### Official Review · Reviewer_oudf · 2021-10-31
**Interesting and simple idea, convincing results and well written paper**

**Rating:** 8
**Confidence:** 4

**Review:**

This article proposes a convolutional network architecture to address the lack of connectivity between features of spatially distant locations within a layer. The authors propose CycleNet, which consists of the concatenation of convolutional operations on the three pairs dimensions - (x, y), (x, z) and (y, z) - instead of only on (x, y). The paper studies several properties of CycleNet compared to some baselines models: the performance on CIFAR-10, the receptive field size of the learnt features and the performance on the Pathfinder challenge.

This is a well written paper, which presents a simple and reasonable idea to address a weakness of standard convolutional models - the lack of connectivity between distant pixels or features. While the analysis of the proposed architecture does not outperform standard models on image classification tasks, the performance is close enough and, importantly, the experiments show the advantageous properties of CycleNet on other dimensions beyond classification accuracy, such as the receptive field size of the features and the performance on other tasks such as Pathfinder. I think the choice of experiments is sound and extensive enough for a workshop submission. Therefore, I have a generally positive impression of this paper and I recommend its acceptance to the SVRHM 2021.

Nonetheless, I have a few comments about potential weakness or aspects that could be improved, as well as some questions. First, I believe that the paper should more transparently present the less positive results of CycleNet from the experimental setup. For example, the authors show the performance on CIFAR-10 compared to a basic CNN baseline in Figure 3 of the main body of the paper, but leave for the supplementary material the results on ImageNet, where CycleNet achieve comparably worse classification accuracy. I argue that this introduces an analytical bias that can be misleading. Second, I think the paper could be improved by more in-depth discussion of the limitations of the proposal and directions for future work. Finally, I would also have appreciated a longer discussion on what the gap is that this new architecture aims to fill if the issue it addresses can be mitigated or solved by architectures such as transformers. I encourage the authors to consider these changes for their camera-ready version, if the paper is accepted.

---

> ### Author Response · Authors · 2021-12-06
> **ImageNet, architecture scaling and future work**
>
> We thank the reviewer for the useful feedback. In the final version of the paper, we will make sure to add ImageNet to the main body of the paper. This was initially moved to the Appendix only due to the hard limit on the number of pages of the submission. The main limitation of the architecture is the scaling of the architecture with the input pixel size due to the nature of the orthogonal convolutions. Future work includes how to decompose tensors in the most effective way, in fact, our work adds also to the recent direction of research decomposing standard operators from convolution to attention and now coming to the decomposition of fully connected layers (see Hanxiao Liu et al. - Pay Attention to MLPs)

---

### Decision · Program_Chairs · 2021-11-02

Accept (Poster)